# Peter IV of Aragon (1336–1387)

**Marta Serrano-Coll** 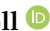

Department of History and Art History, Universitat Rovira i Virgili, 43003 Tarragona, Spain;
marta.serrano@urv.cat

**Definition:** Peter IV king of Aragón (1336–1387). He was the seventh king of the Crown of Aragon, and father of Juan I (1387–1396) and Martín I (1396–1410), the last members of the dynasty to take the throne. When Martín died, the Trastámara branch occupied the throne of the kingdom. Peter IV was dazzling in his ability to use art as a tool of authority and sovereignty. With the aim of exalting the dynasty, he patronised various enterprises, among the most important of which was the abbey of Santa Maria de Poblet, which he intended to be a burial place for himself and his successors, a wish that was fulfilled, without exception, down to Juan II, the predecessor of the Catholic Monarchs. A perfectionist and zealot, he endowed important religious events with profound political significance, and promoted works of great symbolism such as the genealogy of the new *saló del tinell*, or the *ordinacions de la casa i cort*, to which he added an appendix establishing how the kings of Aragon were to be crowned.

**Keywords:** royal images; royal iconography; kings of Aragon; Crown of Aragon; Peter IV of Aragon





## 1. An Exceptional Reign

On the death of Alphonse IV (1327–1336) in Barcelona, the kingdom of Aragon passed into the hands of the prince Peter, nicknamed the Ceremonious for his interest in the due magnificence of the institution he represented and in the palatine entourage, which he organised with care and attention to detail.

He was born on 5 September 1319 in Balaguer. No one foresaw that he would attain the crown: he was the second son and the kingdom then belonged to his uncle James, the first son of James II (1291–1327). James's renunciation of the throne meant that Alphonse, Count of Urgell and Peter's father, became the rightful heir, and this, added to the death of Alphonse's first-born son shortly afterwards, meant that Peter became the legitimate successor. Being in his seventies, he was so weak in health and physically puny that, as he would say in the *Crònica de Pere el Cerimoniós* written in his own hand and which covers his entire reign and that of his father, King Alphonse IV: "neither the midwives, nor those who attended our birth, thought we could live" [1].

Energetic and strong-willed, he increased the power of the monarchical institution, intervened in important foreign conflicts, and extended his dominions by incorporating Sicily, seizing Roussillon and dispossessing the Mallorcan king Jaume III (1324–1349) of his island kingdom. The chronicler Zurita summed up his complex personality: "While this prince was of the weakest and most delicate composure of body, he was also of the most ardent spirit and of incredible promptness and liveliness and of great vigour and execution in all that he undertook, and of spirit and courage for any undertaking and strangely ambitious and haughty and very ceremonious in preserving the royal authority and pre-eminence" [2].

He compared himself to James I (1213–1276), whom he admired with fervour. He considered that he shared several similarities with his predecessor, such as the protection of Providence, similar military exploits and certain biographical facts. His reign was the second longest reign of the Crown of Aragon, after that of the acclaimed Conqueror.

## 2. Cultural Personality: Profitability of Arts and Literature

It is difficult to summarise the artistic and literary policy of this king who, in 1380, praised the Acropolis of Athens as "the most beautiful jewel in the world, such as that not even all the Christian kings put together could build anything like it" [3]. His sensitivity to art went beyond mere delight; he was aware of its value for displaying power. To this end, he promoted works of architecture, among them the *saló del tinell*, the function room of his Palace in Barcelona; the *Palau Menor* for the queen; the restoration of Santa Maria del Mar, where he reinstalled the keystone representing his father Alphonse IV; and his intervention in Poblet, designated the royal pantheon of the dynasty.

He was interested in astronomy, poetry, history and law, and in Arabic, Hebrew and Christian knowledge. He commissioned the translation of notable original works and founded the universities of Perpignan in 1350 and Huesca in 1358. He tried to revive troubadour poetry, writing in a Catalanised form of Provençal, although the most interesting documents are his speeches and personal letters kept in the royal archive, which he rigorously organised (about Peter IV, see: [4–8]).

## 3. Coronation Ceremony and Iconographic Echoes

When he was eight years old, in April 1328, he attended the coronation of his father, defined in his chronicle as "one of the notable festivities that took place in the House of Aragon" (See [1]; Prologue). Its solemnity also impressed the chronicler Muntaner, who described the lavish parade, the procession and the insignia of gold, pearls and precious stones, and the banquet that followed [9]. It was then that the *infante* Peter recited a sirventese he himself had composed on the allegorical interpretation of the insignia: already at such a young age, the man who was to become king showed his predilection for ceremony and the formulas and instruments of protocol.

Peter IV decided to be crowned in Saragossa along "with that trousseau which belongs to a king who is to take coronation" (See [1]; Chapter 2, par. 9). It is significant that he uses the verb "to take" instead of "to receive" the latter implying a passive attitude on the part of the king in deference to the officiating metropolitan. The ceremony took place in the Seo on Easter Sunday 1336, with the Aljafería once again bearing witness to the festivities, in which some 10,000 diners took part in the main meal. He was 16 years old, although his youth did not prevent him from showing his courage when the archbishop, Pedro López de Luna, insisted before the ceremony that he would place the crown on Peter's head, an assertion that caused an argument in the sacristy and delayed the start of the ceremony. The future king did not like the idea; he knew that the exercise of power required symbolic practices, hence his wish to replicate the gesture of his father, who had crowned himself. Consequently, on the advice of his godfather Ot de Montcada, he was determined to demonstrate his authority and independence before the church. He led the cleric to believe that he agreed, but when the time came, he put on the crown himself, telling the prelate "not to adjust it or touch our crown, that we would do that for him" (See [1]; Chapter 2, par. 12, [10–12], more specifically [13]).

In 1353 he ordered a coronation ceremonial to be drawn up so that those who "reign after us would know and, having it written down, could not neglect or not know" [14] how the liturgy was to be performed. He regulated and consolidated each phase of the rite to reinforce the potency of the self-coronation gesture (which displayed his power in the sight of his subjects) and to confirm his idea of separation between the two parts of the ceremony: the spiritual, with the anointing of the archbishop; and the temporal, the coronation, in which the king alone took the insignia directly from the altar. Thus, by eliminating everything that hindered the image of its sovereignty, the monarchy's autocratic character became evident and can be seen in the miniatures of two of the three surviving illuminated codices: the twin copies in the Lázaro Galdiano Foundation and the National Library of France (Ms Reg 14425 and Ms Esp 99, respectively) [15,16] (Figure 1). In them, the sovereign, wearing a paly dalmatic, facing the altar and in front of individuals seated on wooden benches, holds a large fleur-de-lis crown in front of the metropolitan

of Saragossa, whose role is limited to imparting the blessing. Visually, Peter IV shows the elimination of all ecclesiastical involvement in the presentation of the crown which, in the eyes of the people, symbolised the handing over of the kingdom. The same was true of the coronation of the queen who, kneeling and praying, in the three copies receives the crown from the king's hands [17]. The copy from the Fundación March shows the sovereign kneeling, receiving the crown from the metropolitan and in the presence of several prelates [16].

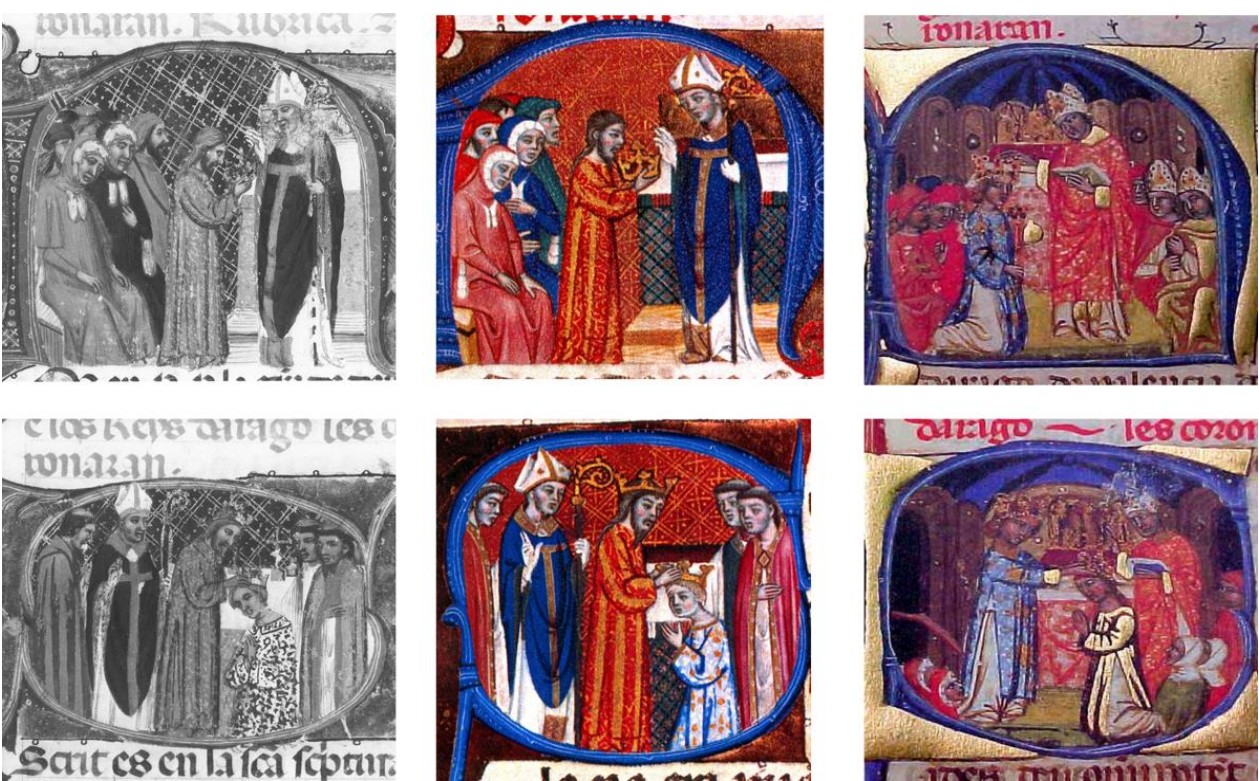

**Figure 1.** *Ceremonial de consagración y coronación de los reyes y reinas de Aragón.* Left: Ms. Esp. 99, Paris, Biblothèque Nationale de France ©; Center: Ms. Reg 14.425, Madrid, Fundación Lázaro Galdiano ©; Right: Ms. Phillips 2633, Palma de Mallorca, Fundación Bartolomé March ©. Images published in [18].

## 4. Conquest of Mallorca: Iconographic Echoes

One of Peter IV's first wishes was to annex Mallorca, at that time ruled by Jaume III. James I had separated it off, along with the territories of southern France, when he ceded them in his will to his second son. From then on, relations between the respective kings were not cordial, as the Aragonese constantly sought to reincorporate it into their dominions. After a previous attempt by Alphonse III in 1285, the Ceremonious finally achieved this and was crowned in Mallorca Cathedral in May 1344. Shortly afterwards on 25 October 1349, Jaume III lost his life in the Battle of Lluchmajor and his body was banished to Valencia Cathedral. The conquest would have iconographic consequences [18].

### 4.1. Coinage

The first area in which these consequences became evident is in his coinage. He sought to integrate his territories economically and one way of doing this was by minting *diners* and *òbols* that followed the traditional model with some changes to his clothing. Specifically he decreed "that [his coins] should again be minted in a similar shape, size and law to the silver coin of Barcelona. And that there should be no diversity except in the lettering" [19] (Figure 2).

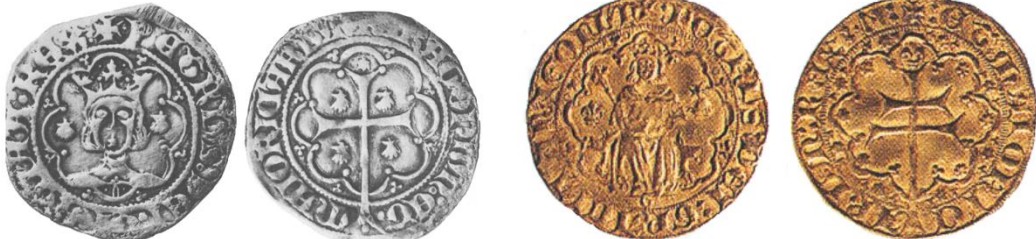

**Figure 2.** *Ral* and *Ral d'or* of Peter IV. Mallorca. Published by Crusafont, *Numimsmàtica de la Corona Catalano-Aragonesa medieval (785–1516)*; Vico: Madrid, Spain, 1982; num. 252 and *Guia art gòtic*, Museu Nacional d'Art de Catalunya: Barcelona, Spain, 1998, p. 54, fig. 1.1.

He then changed his mind and struck, for the first time in the history of the coinage of the kings of Aragon, pieces with a frontal bust/cross and enthroned king/double-barred cross surrounded by + PETRUS DEI GRACIA REX/ + ARAGONUM ET MAIORICARUM, a brief inscription due to the limited size of the pieces and which omitted his sovereignty over the counties of Roussillon, Cerdanya and Montpellier, sold by the Mallorcan king to the king of France to finance the war [20]. Peter IV took special care to mark the invalidity of the sale: COMESQUE BARCHINONE ROSSILIONIS ET CERITANIE would appear on all his seals. These new pieces, precious and full of detail, offered another unprecedented element in the coinage of the kings, namely, mint marks, which were prolific in Mallorcan issues and which from now on would be standard on Aragonese pieces [21].

*4.2. Sigillography*

There are three stages in the changes to the legends and iconographies on Peter IV's stamps and seals [22]. The greatest differences can be seen in the second stage when, in documents from 1343 and 1344 and in accordance with the order that he alludes to in his chronicle (See [1], chap. 3), the king used imprints that were new in iconographic and formal terms, although, typologically, they retain the traditional style: that is, an enthroned effigy on the obverse and an equestrian effigy on the reverse.

In his main seal, the majesty effigy stands out for the change in the design and the complex solium which, surmounted by lions and covered by embroidered cloth, is reminiscent of those in vogue in France from the times of Louis IX [23] (Figure 3). The equestrian image also introduces novelties: first, although it maintains the star preceding the rider, there is a change in the direction in which the king rides, who now shows his right side in accordance with the Anglo-French type [24] and the model used by the King of Mallorca. Secondly, on the head of the Sovereign, which stands out against the filigree background, is the crest of the dragon, this being the first graphic evidence of its use by a king of Aragon (See [17], pp. 91–92).

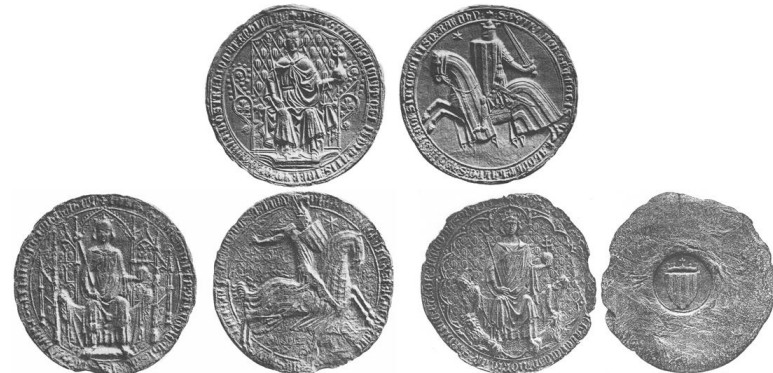

**Figure 3.** Seals of Peter IV. Above: 1337. Below: 1343–1344 and 1344. Published by De Sagarra, *Sigil·lografia*, nums. 57, 59, 58.

In 1344 he issued a new seal also based on Mallorcan pieces (See [22], num. 261). This new type continued without interruption until the Catholic Monarchs and shows the enthroned king and a counterstamp with a shield crowned with the arms of Aragon.

After proclaiming himself king of Mallorca, Peter IV took the iconography of the Mallorcan royal sigillography and, after mixing it with other designs from Aragon and Navarre, created new types exclusively for himself (More details in [18]). He achieved a suitable image by not reusing the same imprints as his deposed predecessor, while at the same time integrating himself into the artistic trends of the time.

### 4.3. Court and Household Ordinances

Encouraged by those close to him [25] and moved by his desire to provide his court with regulations that would guarantee institutional decorum, he ordered the drafting of ordinances for his house and court, the oldest manuscript of which contains annotations in his own handwriting (See [14], estudi introductori). There were earlier ordinances that James II had put into practice after his return from Sicily, perhaps also using certain Hohenstaufen provisions as a model [26,27]. It is no coincidence that Peter IV's date is from 1344, just after the conquest of Mallorca. He seems to have taken the *Leges Palatinae* of the island as a model, given the textual and iconographic coincidences between both manuscripts [28,29].

In his miniatures, the king, elegantly attired, attends to each official whose ordinances he heads. The first folio is extraordinary (Figure 4). The border contains plant and animal motifs and the coat of arms of Aragon (the paly), the cross of Saint George (whom Peter emphasised for his role as defender of the monarchy and of whom he obtained relics) and the cross of Iñigo Arista (which he defined as the "ancient sign of the king of Aragon") [30]. Within the initial, Peter IV sits on a magnificent Gothic throne raised by steps and decorated with coats of arms bearing the pales of Aragon, a pattern that is repeated in the solemn vestments and cushions, including the one beneath his feet. It is a visual representation of what is laid down in the ordinances, both in terms of how the king should dress, his *regalia* and accessories, and in terms of how the attendants should position themselves in relation to him and address him (See [14], among other chaps. 81. De les vestedures e altres ornaments; and 89. De la manera de seer e proposar en consell nostre; Study of its illuminations in [17], pp. 276–283).

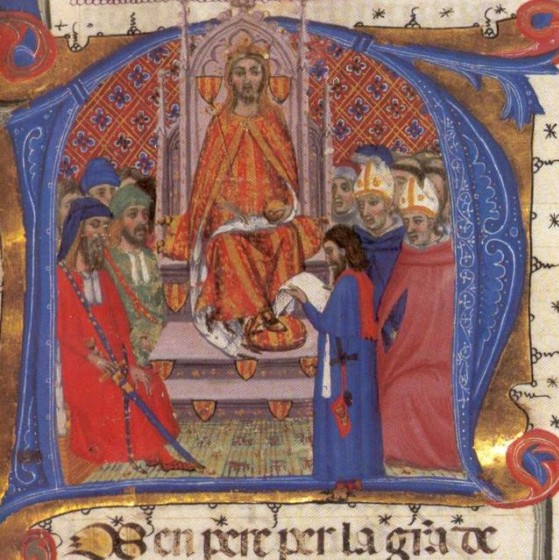

**Figure 4.** *Ordinacions de Cort*, Ms. Esp. 99, Paris, Biblothèque Nationale de France ©. Fol. 1r. Peter IV. c. 1370–1380. Published by Alturo, J. *El llibre manuscrit a Catalunya. Orígens I esplendor*; Generalitat de Catalunya: Barcelona, Spain, 2009, p. 253.

## 5. Artistic Commissions for the Glory of the Monarchy

It is not possible to analyse all the effigies of Peter IV that have survived from the Middle Ages, such as the mural paintings of Daroca, the dozens of illuminations of the copies of the *Usatges i Constitucions de Catalunya*, in the various *fueros*, the *Tercer Llibre Verd*, or in privileges such as those of the Carthusian monastery of Valldecrist. And in sculpture, such as the sepulchre of Lope Fernández de Luna, the reliquary of the corporals of Daroca, or the ephemeral votive offerings, to cite just a few examples (see [17]). But I will detail those commissions that, on a royal and symbolic level, were the most eloquent or significant.

### 5.1. Devotion and Profitability

There were many devotional commissions that he sought to benefit from. For example, his obsession with siring a son to guarantee the continuity of his lineage led him to commission, sometime around 1341–1342, the Book of Hours of Mary of Navarre. Although its pages do not feature effigies of the king, it was commissioned for praying and exhorative purposes, as is illustrated on folio 15v, which shows the young queen praying in front of a *Virgo lactans* [31].

5.1.1. Second Translation of the Relics of Saint Eulalia

Described in detail in the documentation ([32], also see [1], Chap. 2), this translation had iconographic repercussions (Figure 5) and consisted of offering a new burial place to the saint at a particularly propitious moment for the king, who was in Barcelona on one of his first visits as sovereign to receive homage from the man who would shortly afterwards lose his kingdom, Jaume III of Mallorca [33–35].

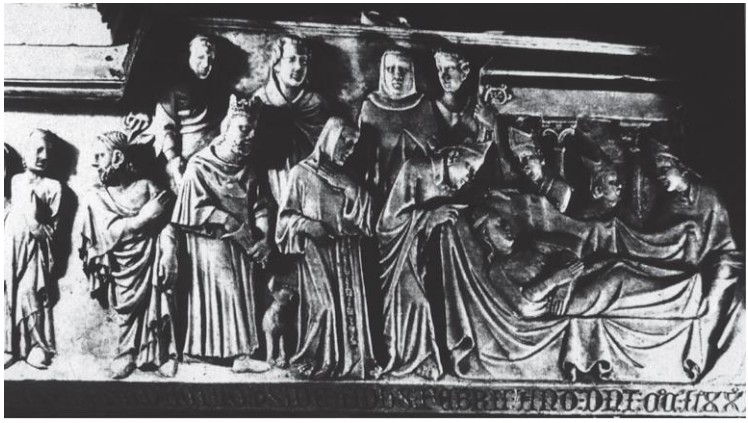 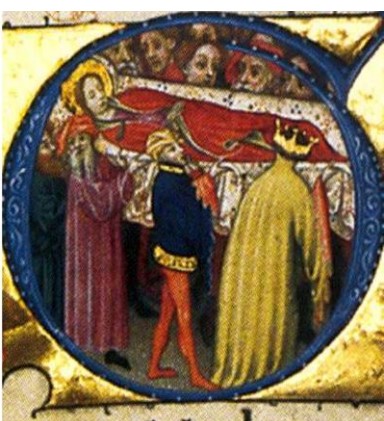

**Figure 5.** Santa Eulalia sarcophagus, detail. c. 1327. © Barcelona Cathedral, serie 28, n. 40; *Misal de Santa Eulalia*, Ms. 116, Barcelona, Arxiu Capitular ©. Fol. 308r. 1403. Published by Alturo, *Llibre ma-nuscrit*, p. 173.

The tomb of the saint follows Italian models by Lupo di Francesco [36]. The cover, by another artist, displays sculpted motifs that were contemporary to the time of its execution, which itself was exceptional in the sculpture carried out in the Crown of Aragon at that time. During the solemn translation of 1339, among the retinue following the religious dignitaries (the Cardinal of Rhodes, the Archbishop of Tarragona and some of the "bishops and prelates of our kingdoms" (See [1], Chapter 2, pp. 34–35), three individuals stand out in the foreground: the one in the centre is Peter IV, who takes off one of his gloves to touch the saint with his right hand before she is placed in the final tomb. To his right is the *Infante* don Jaime, brother of the king and then Count of Urgell [37]. It is unanimously agreed that the king of Mallorca is absent, although he could be the personage also wearing a diadem in the background, just behind Peter IV (See [16], p. 437). Finally, preceding the king, wearing a nun's habit and with a troubled gesture, is Queen Elisenda, by then already living in seclusion at Santa Maria de Pedralbes.

A later iconographic echo is found on fol. 308v of the *Missal de Santa Eulàlia*, illuminated in 1403 by Rafael Destorrents and members of his workshop [38]. The Proper of Saints describes the first translation, but the presence of the king with his back turned (which can be explained by the fact that he had died when the manuscript was completed) seems to indicate that the scene refers to the second translation (See [38] p. 51. Also Planas, J. La Miniatura Catalana del Periodo Internacional. Primera Generación. Universitat de Barcelona: Barcelona, Spain, 1992; p. 441).

5.1.2. The Pantheon of Poblet

Peter IV soon turned his attention to what was to become the dynastic pantheon of the kings of Aragon. The place he chose was the Cistercian monastery of Poblet, breaking with his forebears' preference for the Franciscans. The initiative would be accompanied by other emblematic developments, given that "our monastery of Poblet [...] is the custodian of the bones of the most glorious kings that ever were in the house of Aragon" (See Bracons J. 'Operibus monumentorum que fieri facere ordinamus'. L'escultura al servei del Cerimoniós, in [4]: 220). Among these developments were its fortification, the royal chambers and the library, which is presided over by the inscription "*llibreria del rei en Pere III*" and would house writings in memory of the kings buried there for the propagandistic purpose of legitimising the dynasty by evoking its power and glory through chronicles and genealogies, as the *Llibre dels Feyts del rei en Jacme* by Destorrents, from 1343, or the *Genealogia de Poblet*, or *Rotlle genealogic de Poblet*, circa 1409, preserved in the monastery and where the Ceremonious appears with his characteristic dagger [39].

On 2 January 1377, he declared the monastery the pantheon and burial place for himself and his successors without exception, and he ordered his subjects not to swear allegiance to the new king unless he had first arranged to be buried there [40]. He also closely followed the progress of the works, in which he was directly involved (See [40]. Also, Marés, F. Las Tumbas Reales de los Monarcas de Cataluña y Aragón del Monasterio de Santa María de Poblet; Asociación de Bibliófilos: Barcelona, Spain, 1952.; as well as [16], pp. 402–430. and [4], pp. 209–243). (Figure 6) His recumbent was placed towards the *Capella Reial* and shows him dressed in a deacon's robes and holding a dagger in his hands. It is a faithful replication in stone of the body deposited inside, given that the king ordered that he be dressed in the same clothing and insignia that he wore on the day of his coronation, a scrupulous attention to detail that was also evident in other funerary undertakings ([40], p. 996. and [41]).

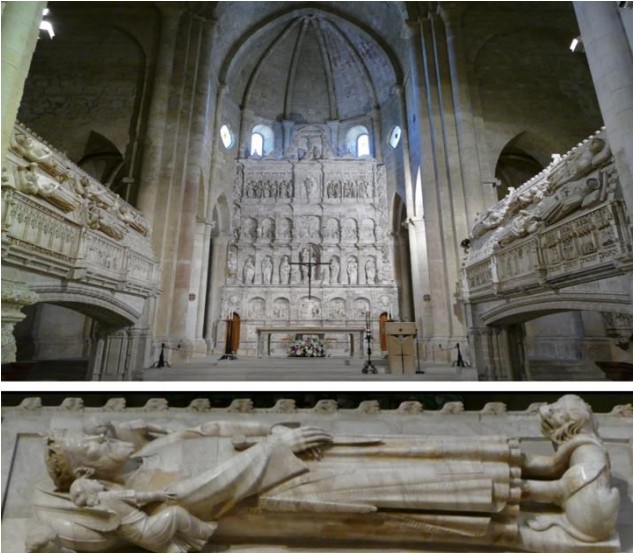

**Figure 6.** *Capella Reial* in Santa Maria de Poblet. Detail of the recumbent figure of Peter IV, restored by Marès. 1944–1949. © Monestir de Poblet.

The pantheon, much remodelled and without the wooden canopies that once crowned it, is also evidence of the political theology of the monarch as *rex et sacerdos* that under Peter IV reached one of its most prominent visual manifestations. What was already patent in his coronation ceremony, in his speeches and in his sermons was also made visible in stone in the pantheon of Poblet through the representation of certain sovereigns such as James I and Alphonse II, who had double recumbents as both kings and as monks [42,43].

*5.2. Other Royal Initiatives: Relevance of Genealogies*

5.2.1. The Saló del Tinell

In 1359 he erected the *cambra major*, or *saló del tinell*, the first stone of which was laid after consulting his astrologers. Peter IV devised the allegorical stories to be painted on the walls, and the commemorative ones above the entrance door [44]. The room was complemented by a gallery of sculpted portraits of members of a political and legal system of the first order, thus legitimising and conferring an aura of glory on them while at the same time giving the room a sense of magnificence. The idea first came to him in 1342 and led him to contract the master Aloi (See [39]: 213–214) to carve 19 statues of kings of Aragon and counts of Barcelona to be attached to the diaphragmatic arches of the hall (See [41], p. 20), the location for of the most illustrious ceremonies. None of these have survived, but their commission is evidence of the monarchy's new urge, under Peter IV, to glorify and perpetuate the dynasty. He wished to show the continuity of his institution by arranging the sculptures of his forebears in order and to inspire the moral rectitude of his successors, who would see in the symbolic hall the images of their exemplary predecessors (See [45], pp. 177–192]. Also [8]). It would also be evocative for his subjects, who would see the representation of dynastic power in an emblematic and performative room to which the most illustrious magnates of the kingdom had access.

5.2.2. The Coronation Sword

In 1360 Peter IV commissioned the Valencian silversmith Pere Bernés, "*fidelis argentarium noster*" (De Dalmases, N. Els argenters de la cort en temps de Pere III. In [4], p. 204), to make a sword with the "most beautiful, richest and most subtle" ornamentation possible. "But in particular we want the outside of the scabbard to feature 19 enamels from one side to the other done in such a way that each one can display the figure of a king or count. Because in these enamels we want to have the figures of the kings of Aragon and counts of Barcelona, past, and our own" [46]. To understand the meaning of this sword, it is essential to relate it to Peter's aforementioned Coronation Ceremonial, which turned the coronation into a visual spectacle (See [45], p. 178). Given the importance of the ceremony, the elements used in it, including the insignia, had to be carefully selected. The promotion of the sword by Peter was probably greatly influenced by his conquest of Mallorca and has been noted by historiography. The king, understanding the ornamentation of this *regalia* as a visual resource and as a categorical mnemonic technique, adopted two important concepts to emphasise and recall the stability of the monarchy over time, namely, lineage and territorial expansion, both of which are identifiable in the iconography of the scabbard [47]. After his annexation of the kingdom of Mallorca, the genealogy represented on the scabbard had a legitimising effect on his conquest. The decoration proclaimed a continuous and uninterrupted dynastic succession, albeit a false one, in the different kingdoms over which the new monarch would rule once crowned. Peter IV was again able to compare himself to James I by linking the sword to the right of conquest, which was the basis of his power, and by handing the sword over to his heirs as an emblem that would play a similar role to that of *Tisó* (*Tizona* or *Tizón* in Spanish is a sword that tradition attributed to Rodrigo Diaz, known by the title of Campeador) in their investitures [48]. This is related to the fact that in the Middle Ages it was common practice to recreate the past within the present in order to legitimise contemporary political practices [49].

## 6. Conclusions

Throughout his long reign, Peter IV was very skilful in the use of art as a tool of authority and sovereignty. With the idea of dynastic exaltation, he promoted the abbey of Santa Maria de Poblet by establishing it as a burial place for himself and all of his successors without exception until the death of Juan II, predecessor of Fernando II the Catholic. Consistent with his project, he endowed the monastery with a royal library that would house historical books commemorating the illustrious dynasty to which he belonged, with walls to safeguard his famous ancestors, royal chambers where he could stay during his stays in the monastery, and a magnificent pantheon that he devised and altered as the works progressed. As a perfectionist, he deliberately sought out those who could provide the tombs with realism and accuracy, an attention to detail that can also be seen in other important undertakings such as the sculptural genealogy destined for the new hall of the *Tinell* or the *ordinacions* which are scattered throughout different libraries in beautifully illuminated codices. In a direct continuation of the practice established by his father Alphonse, Peter IV added to these ordinations an appendix in which he established the manner in which the kings of Aragon were to be crowned, specifying that, during the ceremony, all the insignia, including the crown, could only be handled by the king. This substantial peculiarity, visible to his subjects during the ceremonial, was also represented iconographically in the miniatures that decorate the folios of the ceremonial books except, perhaps tellingly, the *Libro de coronación de los Reyes de Castilla y de Aragón*. That Peter IV conceived art as a means of propaganda is also corroborated by the translation of the relics of Santa Eulalia, on whose sarcophagus the king and perhaps also Queen Elisenda are depicted in the Franciscan habit, an event for which he sought the most religiously and politically advantageous occasion. Above all, there is his conquest of Mallorca, an annexation which, although he considered justified, he legitimised through his iconography by copying without compunction the works of his bitter enemy, the ill-fated Mallorcan king when creating the aforementioned *ordinacions* and the seals and coins that he issued after his accession. It is in relation to this last event that another of his most symbolically important commissions must be attributed, namely, the scabbard of the coronation sword, which showed a genealogy featuring the figures of his predecessors. With this commission, he revived the sword as an insignia with much the same commitment as his much-admired predecessor, James I, whom he had evoked during the conquest of Mallorca and who, by appearing in the iconography of the enamelled royal lineage, also justified it.

**Funding:** This research was funded by *Edificis i Escenaris religiosos medievals a la Corona d'Aragó*, [2017 SGR 1724]. Generalitat de Catalunya-AGAUR.

**Conflicts of Interest:** The author declares no conflict of interest.

**Entry Link on the Encyclopedia Platform:** https://encyclopedia.pub/16717.

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
