# Peer review of "Peter IV of Aragon (1336–1387)"

_encyclopedia, doi:10.3390/encyclopedia1040086_

Round 1

Reviewer 1 Report

The entry on King Pedro IV of Aragon meets the essential characteristics for this encyclopedia. It is very well documented, the critical apparatus is correct, and the illustrations sufficiently clarify the text.

English version:

The encyclopedic entry on Pedro IV of Aragón is based on documentary and bibliographic references of the highest level, both because it includes the most important chronicles that collect the events of King Pedro IV, and the most recent scientific production. The novelty of this contribution lies in analyzing the royal projection of the figure of Pedro IV through art; interesting in that it was reflected in very diverse facets (miniature, seals, sculpture, etc.) highlighting the interest of Pedro IV in establishing a protocol from the moment of the coronation, detaching him from ecclesiastical power. The illustrations that accompany the text contribute decisively to facilitate the understanding of the message by a wide audience.Spanish version:

La entrada enciclopédica sobre Pedro IV de Aragón se sustenta en unas referencias documentales y bibliográficas del máximo nivel, tanto por incluir las más importantes crónicas que recogen los hechos del rey Pedro IV, como por la producción científica más reciente. La novedad de esta aportación reside en analizar la proyección áulica de esta figura a través del arte; interesante por cuanto se reflejó en muy diversas facetas (miniatura, sellos, monedas, escultura, etc.) poniendo de relieve el interés de Pedro IV por fijar un protocolo desde el mismo momento de la coronación desligándolo del poder eclesiástico. Las ilustraciones que acompañan al texto contribuyen decisivamente a facilitar la comprensión del mensaje por un público amplio.

Author Response

Thank you very much for the attention with which you have read my work. Thanks for your time

Reviewer 2 Report

In my view, the article deserves publishing after major modifications which  are mostly structural in nature. The text is very informative and speaks of a well-prepared author, but it has to be reorganized in order to make the argumentation clear for the reader, especially for international readership.

More than anything, the text needs thorough English proofreading. The sentence structure must undergo extensive modifications, first of all, because in its present format, the text looks like a verbatim translation of a Spanish original. Despite the fact that vocabulary issues are relatively rare, I still had to read twice almost every single sentence in order to understand what it was about. Most of the sentences must be shortened (there is a 7-line sentence in lines 215-221), because a word order that works in Spanish does not necessarily work in English. Similarly, the clauses of the numerous compound-complex sentences must be reordered to make the English version truly readable and enjoyable.

The use of pronouns should also be double-checked (e.g. I believe ‘its’ is incorrectly used instead of ‘his’ in line 274); and the art historical terminology at times sounds odd (e.g. I am overall not sure that the word ‘effigy’ is the correct English word choice every time it is used here). Image captions must be translated to English.

Besides the language issues, the text also needs thematic reorganization in terms of essay structure. There is neither introduction nor conclusion; and the title does not reveal what the articles is going to tell about this ruler. As a consequence, I am genuinely not sure what the article was arguing for, even after reading it. It is not a historical summary of Pedro IV of Aragon’s rule, because the narration is almost fully limited to episodes relevant for art. Although one of the keywords is ‘royal iconography’, the article is also not about the iconography of this king. The author says s/he has no intention of analyzing all his representations (line 177-178), and an iconographical analysis should cover the contents of footnote 29.

On the basis of the abstract, my best guess is that the article aims to describe the ruler’s artistic patronage but in its present format, the text looks rather like the description of randomly chosen works of art commissioned by Pedro IV of Aragon. An introduction must be added, where the author tells what s/he will argue for in the article, and a conclusion where s/he summarizes how that point was made.

I also believe that certain expressions in the essay may need further explanation for international readership. These include the frequently quoted primary (or secondary?) sources (at these points, the text of the original source further complicates the Englishness of the article). Furthermore, who is ‘the acclaimed Conqueror’ (line 41)? What is the Crònica dei rei Pere (line 27); the cambra major, or salò del tinell (l. 242)? Who are Zurita (line 32), Muntaner (line 60), and Tisó (l. 279)? “…has been noted” by whom in line 269? These smaller details might be obvious for the author, but less so for international readers less familiar with Pedro IV of Aragon, whom Arts clearly aims to reach. The author must remember that the reader will be less informed than h.self, and the article should be reorganized with that in mind.

I am nevertheless certain that once the language and structural issues are solved, this article will have the potential to become a very valuable publication. Thank you for the chance of reading it, I believe I have encountered an expert author who is better informed than myself and I have learnt a lot. I hope my comments are helpful and not too harsh, offending in any way was not my intention. Should you need any further clarification, please do not hesitate to ask.

Author Response

First of all, thank you very much for the attention with which you have read my work. I will take all your considerations into account.
Once again, thank you very much for your comments, which have undoubtedly improved the result of my work

Round 2

Reviewer 2 Report

Thank you very much for the revised version and apologies for the slight delay in replying.

In my view, the article is now ready for publication.

The text has obviously undergone English proofreading and the modifications have greatly improved readability. Special thanks for the newly inserted clarifications of the referenced sources (l. 30) and names (l. 35 and 60); for the translations and modifications of subtitles (4.3 and 5); as well as for the addition of well-developed Conclusions.

Some typos have remained in the text:

  • l. 10: delete ‘as’
  • l. 42: insert ‘as’ after ‘such’
  • l. 91: insert ‘the’ before ‘monarchy’
  • l. 130: delete ‘which was’
  • l. 131. ‘Pedro’ instead of ‘Pe- dro’
  • l. 169: delete ‘the’ before ‘Pedro IV’
  • l. 294: change ‘Peter’ to ‘Pedro’ for the sake of consistency

A final suggestion: both the abstract (l. 11) and the conclusion (l. 297) mention that the successors of Pedro IV were buried in Santa Maria de Poblet until Juan II - perhaps this latter detail could also be mentioned shortly in the main body of the text. Currently l. 233 only says that SMdP was intended to be a burial place of Pedro IV’s successors without exception (l. 233), giving the impression that indeed all successors were buried there.

Nonetheless, both this detail and the typos are only smaller modifications which take nothing from the overall value of this very illuminating study, now fully able to reach an international audience. Congratulations and thank you very much for the opportunity of contributing to its publication.